# Formal Modeling and Improvement in the Random Path Routing Network Scheme Using Colored Petri Nets

**Muhammad Shoaib Farooq** [1] , **Muhammad Idrees** [2,*], **Attique Ur Rehman** [1,3], **Muhammad Zubair Khan** [1], **Ibrahim Abunadi** [4], **Muhammad Assam** [5] , **Maha M. Althobaiti** [6] and **Fahd N. Al-Wesabi** [7,*]

1    Department of Computer Science, School of System and Technology, University of Management and Technology, Lahore 54000, Pakistan; shoaib.farooq@umt.edu.pk (M.S.F.); F2020279008@umt.edu.pk (M.Z.K.); attique.rehman@lgu.edu.pk (A.U.R.)
2    Department of Computer Science and Engineering, Narowal Campus, University of Engineering and Technology, Lahore 51600, Pakistan
3    Department of Computer Science, Lahore Garrison University, Lahore 54000, Pakistan
4    Department of Information Systems, Prince Sultan University, P.O. Box No. 66833 Rafha Street, Riyadh 11586, Saudi Arabia; iabunade@psu.edu.sa
5    College of Computer Science and Technology, Zhejiang University, Hangzhou 310027, China; assam@zju.edu.cn
6    Department of Computer Science, College of Computing and Information Technology, Taif University, P.O. Box 11099, Taif 21944, Saudi Arabia; maha_m@tu.edu.sa
7    Department of Computer Science, College of Science & Art at Mahayil, King Khalid University, Abha 62529, Saudi Arabia
*    Correspondence: midrees10@uet.edu.pk (M.I.); falwesabi@kku.edu.sa (F.N.A.-W.)

**Abstract:** Wireless sensor networks (WSNs) have been applied in networking devices, and a new problem has emerged called source-location privacy (SLP) in critical security systems. In wireless sensor networks, hiding the location of the source node from the hackers is known as SLP. The WSNs have limited battery capacity and low computational ability. Many state-of-the-art protocols have been proposed to address the SLP problems and other problems such as limited battery capacity and low computational power. One of the popular protocols is random path routing (RPR), and in random path routing, the system keeps sending the message randomly along all the possible paths from a source node to a sink node irrespective of the path's distance. The problem arises when the system keeps sending a message via the longest route, resulting because of high battery usage and computational costs. This research paper presents a novel networking model referred to as calculated random path routing (CRPR). CRPR first calculates the top three shortest paths, and then randomly sends a token to any of the top three shortest calculated paths, ensuring the optimal tradeoff between computational cost and SLP. The proposed methodology includes the formal modeling of the CRPR in Colored Petri Nets. We have validated and verified the CRPR, and the results depict the optimal tradeoff.

**Keywords:** wireless sensor network; source-location privacy; calculated random path routing; modeling; Petri nets; security system

## 1. Introduction

One of the major WSN deployment problems is source-location privacy (SPL) [1]. SLP includes the hiding of all source nodes in the network from attackers [1]. All the nodes are present in an open environment, and all the nodes are sending tokens from a source node to a sink node, so if attackers locate the source node, then the source node is at high risk. Providing privacy in sensor networks is very complicated because the nodes have low battery capacity and less computational power [2]. WSNs are deployed in a wide range of domains, such as academia and industry. Lately, the SLP problem has emerged as a

significant issue in WSNs, especially in critical privacy systems. There is a need to address this problem to make the WSNs secure for transmitting messages [3].

Previous studies have proposed numerous protocols to address SPL, and one of the fundamental protocols is random path routing (RPR) [4]. RPR is used to protect tokens from malicious attackers in the adversary [5]. RPR ensures SLP, however, it is expensive when it starts sending tickets along the longest path each time.

The proposed calculated random path routing (CRPR) model provides significantly better results in token routing. The proposed module's distance calculation is our contribution. The system first calculates the distances of all available paths, and then randomly routes the token to any of the top three shortest paths. In other words, the model first calculates all the reaches of the known way and determines the shortest path, ensuring the SLP has a low computational cost and lower battery consumption. Additionally, it includes the formal modeling and verification of the CRPR, which authenticates the best tradeoff between computational cost and source-location privacy. The platform used for formal modeling is Colored Petri nets.

There are four sections in this research paper. The first section presents the related work in which the strengths and weaknesses of the three routing protocols are discussed. The second section of the paper explains the proposed model CRPR. In the third section, this paper outlines the results of the CRPR model. Finally, the last section presents the discussion and future work.

## 2. Related Work

Wireless sensor networks (WSN) are used to monitor pressure, temperature, sound, etc. [6]. It is also used to monitor the various environmental conditions and physical assets under monitoring and tracking [7].

### 2.1. Background Literature

In monitoring applications, the detection of assets is carried out by the nodes. Whenever the assets are detected, the node becomes the source node, and it starts transmitting packets to a sink node, indicating that the assets have been detected in its surroundings [8]. Usually, the distance between the source node and the sink node is not in the transmission range of sensor nodes. For this reason, WSNs create multi-hop communication [9]. One of the significant issues in WSNs is the security of the packets being transmitted in multi-hop touch.

One major factor is energy, and it is primarily a limited resource in WSNs, so it needs to be carefully designed. Mobile nodes require different protocols than those that are static.

### 2.2. Source-Location Privacy Addressing Scheme

The problem of SLP was introduced by [10]. It has been further discussed in many schemes and system models [11]. The prevention of adversaries from backtracking to the source location is made possible by the routing schemes. Routing schemes also make possible the monitoring and analysis of the WSNs.

#### 2.2.1. Fake Source Routing

The baseline of fake source routing (FSR) was introduced by [12]. It is one of the very first schemes for source-location privacy in WSNs. FSR works by using a set of fake source nodes as a natural source, which can act as a decoy. The mechanism of a fake source is that it generates fake nodes to engineer the network traffic to confuse an adversary that these fake packets are actual packets. Fake packets are encrypted and have the same length, making it difficult for the adversary to differentiate between genuine and fake packets. These fake nodes are carefully positioned to avoid leading adversaries towards the actual source [13]. Two strategies exist that form the baseline for fake source routing. (1) persistent fake source routing, and (2) short-lived fake source routing [14,15]. Short-lived fake source routing is an injection strategy, and it does not require any additional overhead.

This scheme is easy to implement, but it provides poor privacy levels, as the fake sources are short-lived. A fake packet guides an adversary in the direction where there is no source code and makes it easy for the adversary to catch basic packages. A persistent fake routing scheme is also similar to a short-fake source, but the point is that only one artificial node is not enough to distract an adversary. So, for this reason, the constant fake source, once it decides to become a phony source, regularly generates bogus packets so that the adversary can effectively be determined.

### 2.2.2. Shortest-Path Routing (SPR)

SPR was also introduced for source-location privacy [14]. In shortest-path routing, a single path is carried out between the source and a destination node. In shortest-path routing, the packet has the shortest distance to travel, and in this scheme, packets are always moving to the next-hop node. Packets are moving along the shortest path between the sink node and the source note. It consumes very little energy, and the delivery ratio with shortest-path routing is very high but provides very low source-location privacy. Additionally, the drawback of this scheme is that it uses network configuration, which is not possible in real-world scenarios.

### 2.2.3. Dynamic Fake Source Routing

Dynamic fake source routing is also a distributed solution of fake source routing called, phantom source routing [16]. There are many paths available for the source node to send data to the sink node, and the data can be sent to the sink node from every possible path. Source-location privacy (SLP) is essential in WSNs as, in most cases, the source node can be exposed to diversity [17]. Fake random path routing provides a significant source of location privacy, but it is too expensive computationally [18]. The information sent from node to sink is exposed to the cyber city and the source's location, which needs source-location privacy [19]. One of the possible solutions for source-location privacy is a random path routing protocol [19].

This research paper proposes an improvement to the random path routing scheme to remove its deficiencies by adding the novelty factor of distance calculation of paths. Furthermore, a Petri net-based formal modeling of the improved routing scheme is proposed, which leads to verifying and validating our proposed routing scheme CRPR. The proposed technique is based on the strategy that the random selection of the path will be made from the top three shortest paths. Most research papers have focused on source-location privacy and ignored the computational cost [20]. While working with source-location privacy, all the nodes in geographically different locations have limited memory, energy, and computational capacity [21]. So, establishing an optimal tradeoff between source-location privacy and computational cost has made the system more secure and efficient. Many protocols have been proposed to address SPL, and one of the fundamental protocols is random path routing (RPR). RPR is used in the adversary to protect tokens from malicious attackers [16]. Although RPR ensures SLP, it becomes costly when it starts sending tokens along the longest path each time.

## 3. Material and Methods

Simulation on Colored Petri nets is carried out with the help of formal modeling before their actual installations are carried out [14]. With the help of this technique, the behavior of the system is shown. It is observed that the significant advantages of using formal modeling are that it shows all the rules and regulations that are implemented in the existing system. One cannot ignore the importance of formal modeling in engineering structures [18]. Before the installation of the system, proper confirmation is required that shows the approaches and objectives. It is necessary to remove the ambiguity while working with the security-based structure, and formal modeling can easily do this. Numerical or mathematical complex problems can also be structured by formal methods [22]. Logically

complex structures can also be structured more comprehensively by using formal modeling techniques. It also provides the facility of verification and validation.

Below, Figure 1 shows the proposed methodology of the model. The CPN tool is used to formally model in the model. The token from a source node to the destination node is sent by using calculated random path routing. First, the model has to find all the possible available paths to send the packet from source to destination. The model has traversed all the possible nodes of that path and gets the calculated path distance. Before sending the packet to the network, the model has all the routes from the source node to the sink node. Now the model has the choice to send the packet from either the shortest path or longest path. It depends upon the token to be sent. The model can send the message from the shortest path again and again if the message does not contain critical information.

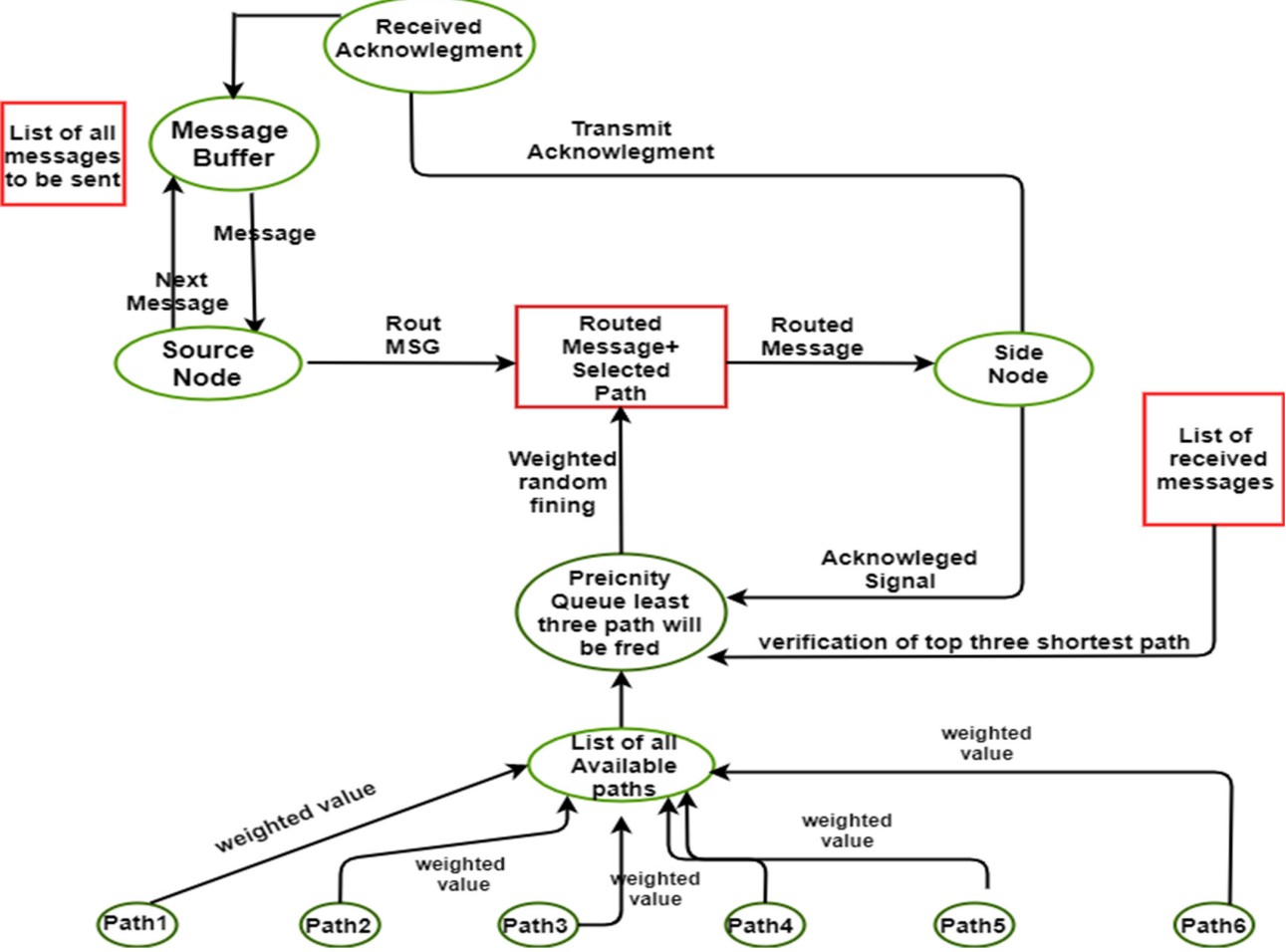

**Figure 1.** Proposed model.

WSN is used to monitor sound, temperature, pressure, etc. Most source nodes are exposed to the cyber city, so source-location privacy (SLP) is necessary [23]. Separate path routing or multi-path routing are protocols that are used in SLP [24]. Separate path routing or multi-path routing is used to secure the location of the source node while sending the message from a source node to a sink node, but it is computationally expensive. To reduce the computational cost, the proposed modification of the current working protocol adds an additional feature is being added in this research to reduce the computational cost. SPR is computationally expensive when the longest path is selected to transfer data from a source node to a sink node. This computational cost can be minimized by first calculating the distances of all paths. After then selecting the top three shortest distances from all the distances and finally a section of one path from the top three shortest distances will

decrease the computational cost. By doing this the existing protocol will become separate shortest-path routing variables mentioned in Tables 1 and 2.

**Table 1.** Variable declaration for color set.

| | Color Set | Description |
|---|---|---|
| **COL1** | Closet li = list INT; | The color set li defines the List of Integers. The places having the datatype li contains the list |
| **COL2** | Closet PRIOR = list INT; | The color set PRIOR defines the list of integers. The places contain the token values. |
| **COL3** | Color set PRIORITY = INT; | The color set PRIORITY contains the integer values of the tokens. |
| **COL4** | Color set revind = product li * PRIORITY; | The li and PRIORITY belong to the color set the rewind. It contains the values of the product of li and PRIORITY. |

**Table 2.** Variable declaration for system.

| | Variables | Description |
|---|---|---|
| **V1** | Var y:revind | The variable y belongs to the color set the rewind. It holds the values of the rewind color set |
| **V2** | Var p | The variable p holds the values of the priority value. |
| **V3** | Var x:li; | The variable x belongs to the Color set li. It holds the values of all list elements. |

### 3.1. Hieratical Colored Petri Nets

Hieratical Colored Petri nets define the abstract view of the model. It contains the sub-modules of the model. Each module is connected to the other modules and passes its values to the other modules. The hierarchal Petri net of the system is shown in Figure 2.

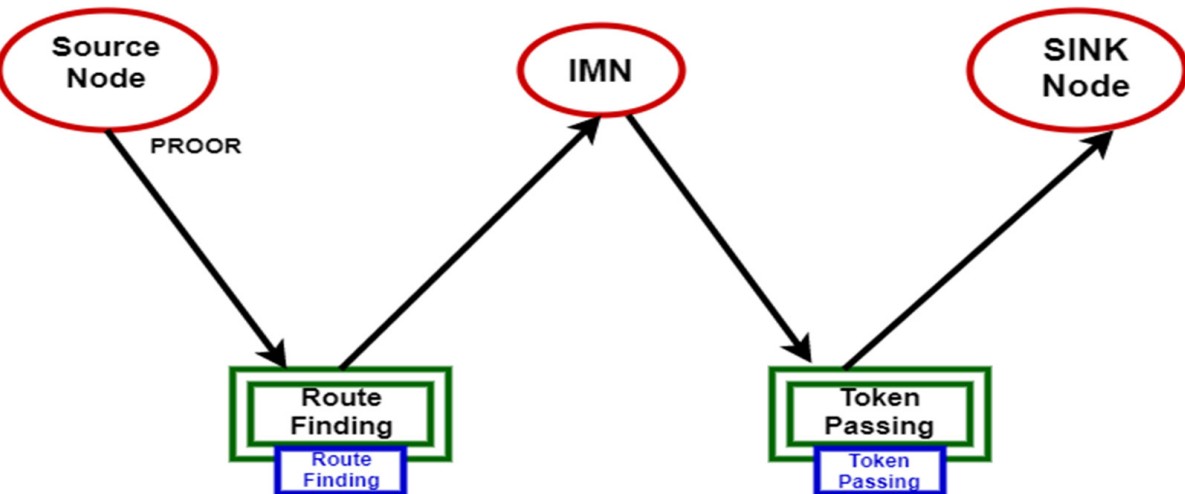

**Figure 2.** Hieratical Colored Petri Nets.

There are two sub-modules in the system.

1. Route finding
2. Token passing

### 3.1.1. Route Finding

The input for the route finding module is the empty list. The route finding module transverses all the routes one by one and calculates the distance between all routes, as shown in Figure 3. It generates a list that contains the calculated distance in integral form.

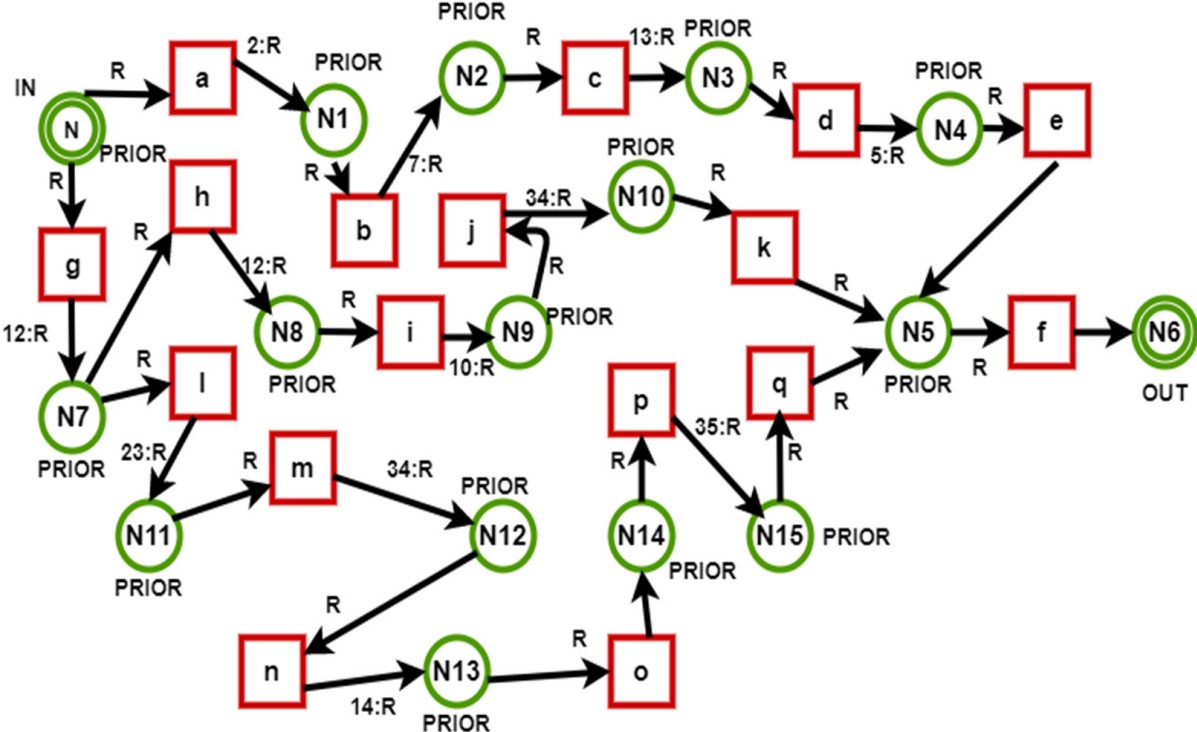

**Figure 3.** Root finding.

Token Passing: The input for the passing token module is the list of calculated paths' distance. The token can be fired to any of the calculated distances, and the distance of the route would be already known.

Figure 4 below shows the transfer of data from a source node to a sink node. Here it can be seen the distance between the nodes. There is more than one path for packets to go from source to destination. A problem arises when the system selects the longest paths again and again, causing an increase in the computational cost.

The random path routing is explained in Figure 4. Figure 4 shows that it has a source node and sink nodes N and N6, respectively. Suppose the token is fired from node N to node N1 by passing through the transition named as "a." As the token fires the transition, it will calculate the distance by the variable R, which is 2: R. This means the distance between the nodes N and N1 is 2. While moving forward from N1 to N2, the weight between the nodes N1 and N2 is 7. The model has added the weight of the following distance to the list. So cumulatively, there are two distances on the list. The distance between N2 and N3 is 13. So now the list values are 2, 7, and 13. Moving forward to the next node list value for N3 to N4 it is now 2, 7, 13, and 5. The model has created a list of all calculated distances from a source node to a sink node. The routes' distances go from a source node to a sink node to fire the token. Moving forward, we have the following two options now: one to send the message from the same route as the system has sent it already, or to send it from any other route. A simple message sent that does not contain the critical information can be sent along the same route, the shortest route. However, if it has critical information to send, the system can send it on its own considering its computational power.

### 3.1.2. Token Passing

There are two equations utilized for token passing shown below as follows:

1.  Pin (path routing) = [] (Pin input for the module, and [] is the empty list)
2.  Pout (calculated distance) = [list of all paths distance] (Pout is the module's output that contains paths distance)

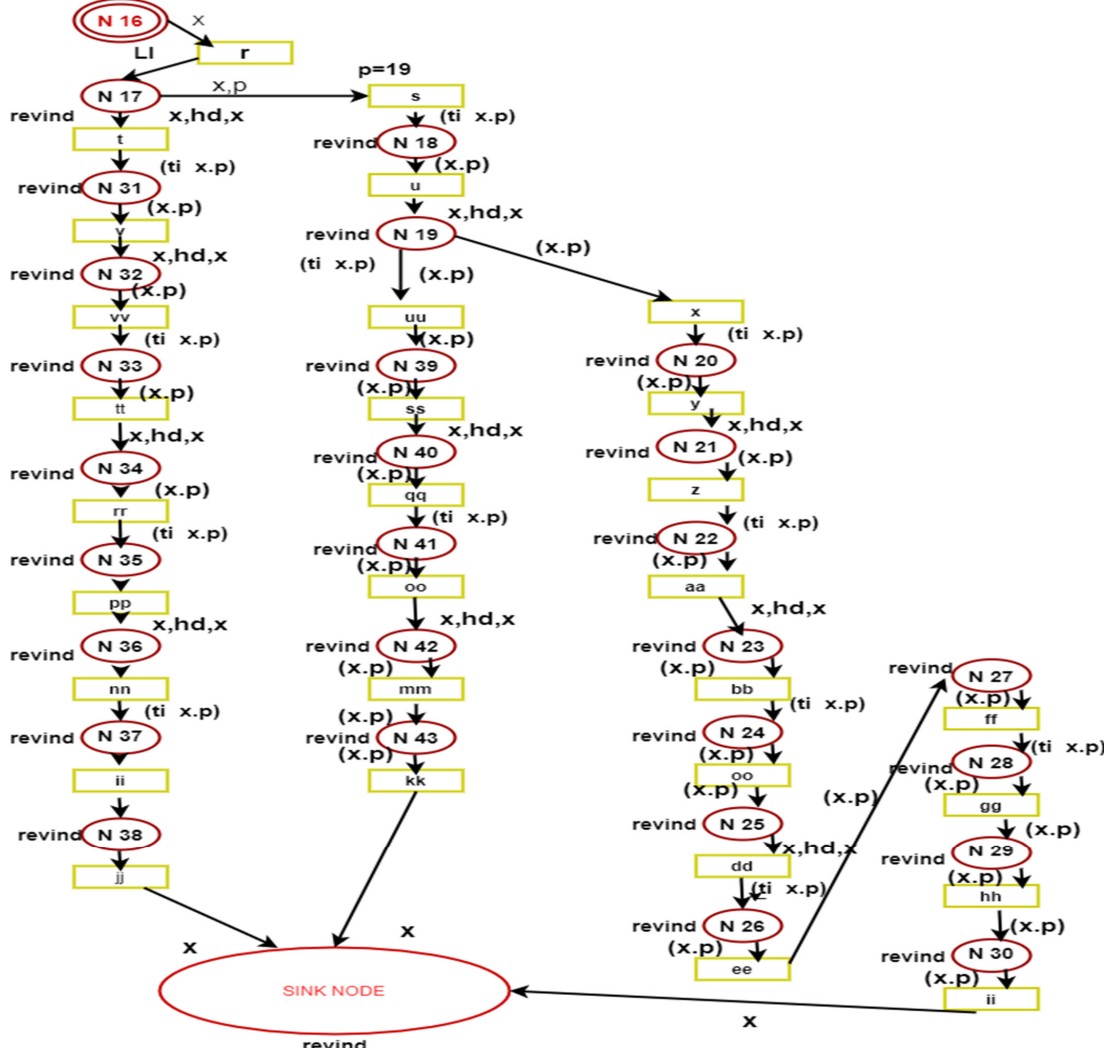

**Figure 4.** Token passing.

The above equations show that the input for this module's route finding is the list of the calculated distances from a source node to a sink node. When the token is passed through the source node to the sink node, it follows the same path provided by the route-finding module in Figure 4, token passing.

In Figure 4 the node, 16, has the calculated path in the form of a list. The data type of Node 16 is li = list. It transfers the list to the N17. Here the reverse of the whole list takes place. Now at the head of the list, which is the following path to be traversed, Node 17 will pass the head value of the list to node N 31. Having had the list whose values are li = {5, 13, 7, and 34}. After reversing this list, the list becomes li = {34, 7, 13, and 5}. The model has calculated the head of the list by the function (x. HD, x). As the token will move forward, the next transition has a guard value of p = 34. The head of the list is where the next transition is processed, and the guard value matches. So, the token is passed along this route. Here again, Node 31 will pass the token to Node 31.

The list value that the list had now becomes li = {5, 13, 7} as the head of the list has been removed. In between Node 31 and Node 32, the same process is repeated. The list is reversed and pass the head value of the list, and now the next token is ready to be sent. After reversing the list becomes li = {7,13,5}. By using a head function, the model has calculated the head of the list.

The guard value of the transition VV is 7, and the head value of the list is the same, so the token is passed from N 31 to N 33. The new value of the list is now li = {5, 13} after the removal of the head value. The head value of the list and the transition's guard value

validate the token to be passed on the right path. If the head value and the guard value are different, the transition is no longer active.

So, the system has passed two nodes by validating the guard value of each transition. Now the list value is li = {5, 13}. The same process is repeated, and the model has reversed the list between Node 33 and Node 34. The list value becomes li = {13,5}. The head of the list is calculated here to validate the head value and the guard value. As the guard value of transition RR is 13 and the head value of the list is 13, the model can move the token from Node 34 to Node 35. After transferring the value to Node 35, the list values become li = {5}. So, for this token, as the model moves forward from Node 35 to Node 36, again, the same process is repeated, and the list is traversed, and it becomes li = {5}. When the system has found the head value of the list, the result is 5. The transition NN has the guard value of 5, so the system has validated, and a token is passed from the transition NN. After the transition is active, the list value becomes li = {}, which means it is empty. This means that this is the end of the path, and the token has reached the destination sink node—the system a calculated specific path and the different routes to be available to send tokens. The system has automatically traversed all the nodes and transitions followed by the list, which shows the validation of the system and path that has been traversed.

## 4. Experimental Results

The results show an improvement in the existing protocol. Our model has successfully calculated all the paths and routed the packets along the calculated path. The system selected the top three distances from all the calculated paths. The verification and validation of CRPR are discussed below.

### 4.1. Verification and Validation

Figure 4 shows the state-space analysis of the model, showing that it has a total of 45 nodes and 50 arcs in the reachability graph, and its status is full. The state-space graph has a total of 32 nodes, and the state-space status is shown as full. This means that there is no such state that is out of the reach of a model. In our scenario, the token that has passed from the source node to the sink node has traversed all the states. This property of validation authenticates our proposed model.

#### 4.1.1. Home Markings

In our model, the initial marking is not a home marking. This means that the initial marking of the model is different from the home marking.

#### 4.1.2. Dead Marking

Verification of the model depends upon the dead markings of the model. In our model, there are three dead markings (21, 35, and 28). Additionally, all three of these nodes are end nodes called sink nodes. This property shows that our model is reaching the endpoint of the network.

#### 4.1.3. Live Transition Instances

Our proposed model displays live transition instances as none, which indicates the optimal efficiency of the model.

#### 4.1.4. Fairness Property

This type of property tells us whether there exists infiniteness in the model or not. In our proposal, there is exists no infiniteness. In short, there is no loop in the whole model.

In Figure 5, the state graph shows the state-space graph of the model. In the state-space graph, all the nodes are linked, which means that all the nodes are traversed. This graph shows the authenticity and validation of our proposed model. CRPR has shown the best results in token passing, and it traversed all the nodes present in the network. It calculated

all the distances from source note to sink node, irrespective of their location, and then traversed the token along any of the top three shortest paths.

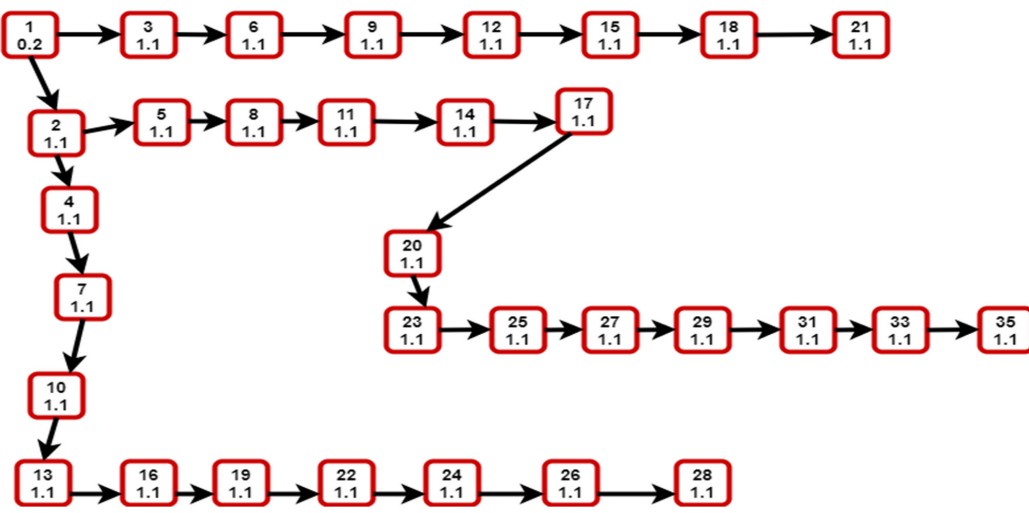

**Figure 5.** State-Space Graph.

### 4.2. Comparison of CRPR with Other Similar Routing Techniques

The comparison of our proposed CRPR model with other similar routing techniques is shown in Table 3. The phantom routing-based techniques are not formally modeled by Petri nets, and it also does not use any guard value. It is a single-path routing scheme, and it can have repetitive packets sent. The fake source-based techniques are also not modeled yet, and it is a multi-path routing technique. Dynamic source routing is a secure network technique, but not formally modeled yet, and it is a multi-path routing technique. Associativity-based routing is also not modeled yet, and it is a multi-path routing scheme. Our proposed technique models the shortest selective random path routing technique and is formally modeled by using the guard technique. SSRPR is a multi-path outing scheme and its reachability is 100%. However, this model does not have a liveness property. The comparison of the proposed model with the existing techniques shows relatively good results so far as the security of the packets is concerned.

**Table 3.** Comparison of CRPR with other similar routing techniques.

| Protocol | Formally Modeled | Guard Checking | Multi-Path Routing | Reachability | Liveness |
|---|---|---|---|---|---|
| Phantom routing-based Techniques | NO | NO | NO | YES | YES |
| Fake sources-based Techniques | NO | NO | YES | YES | YES |
| Dynamic Source routing | NO | YES | YES | NO | YES |
| Associativity- based routing | NO | NO | YES | YES | NO |
| Dynamic backup routes routing protocol | YES | NO | YES | NO | YES |
| Hint-based probabilistic protocol | NO | NO | YES | NO | NO |
| Shortest selective random path routing | YES | YES | YES | YES | YES |

## 5. Conclusions

Source-location privacy is an essential issue in routing protocols. With the invention of new and critical information transmission methods, the safe transmission of data is a critical issue. Routing protocols have managed to solve this problem, but there is always a margin for improvement. The model presented is our best attempt to give positive

input to solve this problem by improving the already existing random path routing protocol and naming it Calculated Random Path Routing (CRPR). The model considers the tradeoff between privacy and computational cost to improve the existing routing protocol. In our future work, we will try to improve this model using computational cost and privacy viewpoints.

**Author Contributions:** Conceptualization, M.S.F.; data curation, M.I.; formal analysis, A.U.R., M.I., M.S.F., M.Z.K. and I.A.; investigation, M.A., M.M.A.; methodology, M.S.F.; project administration, M.S.F. and M.I.; resources, M.S.F. and A.U.R.; software, M.S.F.; supervision, M.S.F., A.U.R. and M.Z.K.; validation, M.I.; visualization, M.M.A.; writing—original draft, M.S.F., A.U.R. and F.N.A.-W.; writing—review and editing, F.N.A.-W., M.I., M.S.F. and A.U.R.; proofreading and writing the paper script in overleaf, M.S.F. and M.Z.K. All authors have read and agreed to the published version of the manuscript.

**Funding:** Taif University Researchers Supporting Project number (TURSP-2020/328), Taif University, Taif, Saudi Arabia.

**Institutional Review Board Statement:** Not Applicable.

**Informed Consent Statement:** Not Applicable.

**Data Availability Statement:** Not Applicable.

**Acknowledgments:** The authors extend their appreciation to the Deanship of Scientific Research at King Khalid University for funding this work under grant number (RGP 1/14/43). We deeply acknowledge Taif University for supporting this research through Taif University Researchers Supporting Project Number (TURSP-2020/328), Taif University, Taif, Saudi Arabia. The authors would like to acknowledge the support of Prince Sultan University for paying the Article Processing Charges (APC) of this publication.

**Conflicts of Interest:** The authors declare no conflict of interest.

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
