# Peer review of "Formal Modeling and Improvement in the Random Path Routing Network Scheme Using Colored Petri Nets"

_applsci, doi:10.3390/app12031426_

Round 1

Reviewer 1 Report

The manuscript proposed a calculated random path routing for transmitting a token  via three shortest routed path to minimize the computational cost while  preserving and ensuring the privacy of the source location. However, the contribution of the research to knowledge is vague as there are so many source location privacy scheme that have been proposed that are built on random path routing among which include

  1. Li, Y., Lightfoot, L., & Ren, J. (2009, June). Routing-based source-location privacy protection in wireless sensor networks. In 2009 IEEE International Conference on Electro/Information Technology (pp. 29-34). IEEE.
  2. Li, Y., Ren, J., & Wu, J. (2011). Quantitative measurement and design of source-location privacy schemes for wireless sensor networks. IEEE Transactions on Parallel and Distributed Systems, 23(7), 1302-1311.
  3. Han, G., Zhou, L., Wang, H., Zhang, W., & Chan, S. (2018). A source location protection protocol based on dynamic routing in WSNs for the Social Internet of Things. Future Generation Computer Systems, 82, 689-697.

Among others. Furthermore, rather than using coloured petri Nets, the formulation of a mathematical model and using optimization techniques to determine the optimal path would have given the work more credence than just assigning weight to the path which may be unrealistic as the network grows bigger. In addition, the manuscript in its current form needs proof reading as it contains so many grammatical and typographical error such as:

  • …This research paper proposed the Calculated Random Path Routing and the novelty (CRPR)…( line 32- 33)
  • ... deployed in may (line 47), SLP and not SPL (line 51), its (line 83), hierarchical not hieratical (line 187) among others

Line 231 and 232 are not standard way of written equation and conveys no information. Also, there was no comparison of their proposed CRPR with other similar routing techniques that determine the shortest path for routing information from source node to the sink toward preserving the SLP

Author Response

Thank you for your feedback on our paper which has greatly helped us to improve the quality and presentation of the paper. We have carefully addressed all the comments raised by the reviewers in this revised version. We explain below how the comments of each reviewer have been carefully addressed. We have highlighted the major changes we have made in the revised manuscript in yellow highlighted color. We sincerely hope you and the reviewers will be satisfied with all the revisions we have made.

Thank you for your consideration.

Sincerely,

Authors

Comments Reviewer#2:

The manuscript proposed a calculated random path routing for transmitting a token via three shortest routed path to minimize the computational cost while preserving and ensuring the privacy of the source location. However, the contribution of the research to knowledge is vague as there are so many source location privacy scheme that have been proposed that are built on random path routing among which include:

  1. Li, Y., Lightfoot, L., & Ren, J. (2009, June). Routing-based source-location privacy protection in wireless sensor networks. In 2009 IEEE International Conference on Electro/Information Technology (pp. 29-34). IEEE.

Author response : We thank the reviewer for the valuable suggestions.

Li et al.  present a randomly selected intermediate node (RRIN) scheme for local source location privacy protection, whereas we provide an optimized graphical modeling technique for source location privacy and also proposed shotest random path routing technique using Color Patri Nets.

  1. Li, Y., Ren, J., & Wu, J. (2011). Quantitative measurement and design of source-location privacy schemes for wireless sensor networks. IEEE Transactions on Parallel and Distributed Systems, 23(7), 1302-1311.

Author response :

Li et al.  present a criteria to quantitatively measure source location privacy for routing-based schemes and focused on message delivery latency, whereas we have provided an optimized graphical modeling technique using Color Perti Nets for source location privacy and our proposed model also considers trade-off between computational cost and source location privacy.

  1. Han, G., Zhou, L., Wang, H., Zhang, W., & Chan, S. (2018). A source location protection protocol based on dynamic routing in WSNs for the Social Internet of Things. Future Generation Computer Systems, 82, 689-697.

Author response :

Han et al.  present a dynamic rounting scheme based on greedy route selection and preserves source location privacy, whereas we have provided an optimized graphical modeling technique using Color Petri Nets for source location privacy and our proposed model also considers trade-off between computational cost and source location privacy. Additionally, Our proposed model calculates all available paths from source to destination and selects the top three shortest paths. The token has be send arbitrarily on any of these three choosen paths. It enusures to reduce computational cost and also make secure source location privay i.e trade-off between computational cost and source  location privacy.

We have also  added the lines 313-323 to differentiate CRPR with other relevant techniques. The newly added lines clearly highlight the contribution of CRPR model.

Furthermore, rather than using coloured petri Nets, the formulation of a mathematical model and using optimization techniques to determine the optimal path would have given the work more credence than just assigning weight to the path which may be unrealistic as the network grows bigger.

Author response: We thank the reviewer for the valuable suggestions.

Color Petri net is graphical and mathematical modeling language used to describe optimal solution of a system. Petri nets are a strong language has been used for specification, analysis, and synthesis of programs and it has been used to represent the complex parallel or concurrent activities in a system efficeintly. We use color Petri nets to make it easier to understand the problem of source location privacy problem to formally model because of its because of their graphical and precise nature of presentation.Petri nets are equally suited for representation of hardware and software modeling problems in  wireless sensor networking systems. We have already defined mathematical varaibles in table1 and table 2.The use of mathematical model and optimization techniques to determine the optimal path can be more realistic however, the assigned weights to the path is applicable in large networks. Nevertheless, in future work we deem to consider this suggestion.

In addition, the manuscript in its current form needs proof reading as it contains so many grammatical and typographical error such as:

  • …This research paper proposed the Calculated Random Path Routing and the novelty (CRPR)…( line 32- 33)
  • ... deployed in may (line 47), SLP and not SPL (line 51), its (line 83), hierarchical not hieratical (line 187) among others

Author response: We agree to the comments of the reviewer and have corrected the grammatical and typographical errors. In particular, we corrected line 32-33, line 47, line 51, line 83 and line 187. The authors have proof read the whole manuscript for corrections.

Also, there was no comparison of their proposed CRPR with other similar routing techniques that determine the shortest path for routing information from source node to the sink toward preserving the SLP

Author response: We thank the reviewer for the valuable suggestions  and have incorporated the suggestion into the manuscript. The authors added a new Table 3 in which, the comparison of CRPR model with other similar routing techniques have been portrayed. Also, the authors added line 313-323 to explain the difference and novelty of CRPR.

Reviewer 2 Report

In general, this manuscript presents a new model CRPR and its tradeoff, which could be important to this field. But the writing style needs to be significantly improved and there are many typos and grammar mistakes. It gives the reader an impression that this manuscript was carelessly organized. For example, line 55- 58 and line 138-142 are EXCATLY the same. It makes people wonder why the authors needed to repeat the sentence word by word. Line 100-101 contains several grammar mistakes. Also, in line 294, "Here are the three Live-ness Properties of our proposed model", this sentence seems incomplete. Line 277, "The verification and Validation of", why the word "Validation" is capitalized here? 

Author Response

Thank you for your feedback on our paper which has greatly helped us to improve the quality and presentation of the paper. We have carefully addressed all the comments raised by the reviewers in this revised version. We explain below how the comments of each reviewer have been carefully addressed. We have highlighted the major changes we have made in the revised manuscript in yellow highlighted color. We sincerely hope you and the reviewers will be satisfied with all the revisions we have made.

Thank you for your consideration.

Sincerely,

Authors

Comments Reviewer#1:

In general, this manuscript presents a new model CRPR and its tradeoff, which could be important to this field. But the writing style needs to be significantly improved and there are many typos and grammar mistakes. It gives the reader an impression that this manuscript was carelessly organized. For example, line 55- 58 and line 138-142 are EXCATLY the same. It makes people wonder why the authors needed to repeat the sentence word by word. Line 100-101 contains several grammar mistakes. Also, in line 294, "Here are the three Live-ness Properties of our proposed model", this sentence seems incomplete. Line 277, "The verification and Validation of", why the word "Validation" is capitalized here

Author response: We thank the reviewer for the valuable suggestions. Suggestions incorporated,  we have carefully revised the overall writing style of the manuscript.  In particular, we removed the lines 138-142 as they were unnecessary. Moreover, we corrected the line 100-101 and improved grammar. Furthermoe, we improved the grammar in line 294 and 277 as well. Finally, the word “Validation” being capitalized was a typo mistake which also revised.

Round 2

Reviewer 2 Report

The language has been revised and the overall quality of the manuscript is improved.